# Evolution and Medical Significance of LU Domain−Containing Proteins

**DOI:** 10.3390/ijms20112760

**Published:** 2019-06-05

**Authors:** Julie Maja Leth, Katrine Zinck Leth-Espensen, Kristian Kølby Kristensen, Anni Kumari, Anne-Marie Lund Winther, Stephen G. Young, Michael Ploug

**Affiliations:** 1Finsen Laboratory, Ole Maaloes Vej 5, Righospitalet, DK-2200 Copenhagen, Denmark; Julie.Maja@finsenlab.dk (J.M.L.); katrine.espensen@finsenlab.dk (K.Z.L.-E.); kristian.kristensen@finsenlab.dk (K.K.K.); Anni.Kumari@finsenlab.dk (A.K.); Anne.Marie@finsenlab.dk (A.-M.L.W.); 2Biotechnology Research Innovation Centre (BRIC), Ole Maaloes Vej 5, University of Copenhagen, DK-2200 Copenhagen, Denmark; 3Department of Medicine, University of California, Los Angeles, Los Angeles, CA 90095, USA; sgyoung@mednet.ucla.edu; 4Department of Human Genetics, University of California, Los Angeles, Los Angeles, CA 90095, USA

**Keywords:** uPAR, snake venom α-neurotoxins, GPIHBP1, plesiotypic disulfide bonds, protein evolution, Ly6/uPAR domains, protein module, protein domain

## Abstract

Proteins containing Ly6/uPAR (LU) domains exhibit very diverse biological functions and have broad taxonomic distributions in eukaryotes. In general, they adopt a characteristic three-fingered folding topology with three long loops projecting from a disulfide-rich globular core. The majority of the members of this protein domain family contain only a single LU domain, which can be secreted, glycolipid anchored, or constitute the extracellular ligand binding domain of type-I membrane proteins. Nonetheless, a few proteins contain multiple LU domains, for example, the urokinase receptor uPAR, C4.4A, and Haldisin. In the current review, we will discuss evolutionary aspects of this protein domain family with special emphasis on variations in their consensus disulfide bond patterns. Furthermore, we will present selected cases where missense mutations in LU domain−containing proteins leads to dysfunctional proteins that are causally linked to genesis of human disease.

## 1. Introduction

Protein domains are autonomous folding units that may function alone or as building blocks in the context of multidomain proteins. When such protein domains are encoded by exons flanked by introns of identical phases, they may become genetically mobile and prone to exon shuffling, resulting in the insertion of a domain into a non-homologous protein environment. This process is facilitated by intronic recombination [1]. Highly mobile protein domains are termed protein modules. Examples of protein modules that occur in multidomain proteins include kringle domains, growth-factor-like domains (GFD), fibronectin type I–III (FN1, FN2, and FN3) domains, immunoglobulin domains (Ig), and complement control protein (CCP) domains [1,2]. Along with single-gene or large-genome duplication events, exon shuffling provides a rich source for the evolutionary diversification and neo-functionalization of a given protein domain. The current review focuses on one such domain—the Ly6/uPAR (LU) protein domain. In an evolutionary context, LU domain proteins occur in a wide range of eukaryotic taxa and come in a variety of different flavors: i) as secreted single domain proteins; ii) as glycosyl-phosphaditylinositol (GPI-anchored) single domains; iii) as GPI-anchored multidomain proteins; and iv), as the extracellular ligand-binding domain in the TGF–β receptor family of transmembrane proteins. In this review, we will predominantly focus on deletions of plesiotypic (ancestral) disulfide bonds and acquisitions of apotypic (non-consensus) disulfide bonds in LU domain–containing proteins and discuss some possible functional consequences thereof. In a functional context, these proteins participate in a diverse array of different biological processes such as fertilization, regulation of complement activity, intravascular lipid metabolism, fibrinolysis, cytokine signaling, envenomation, limb regeneration, embryogenesis, and morphogenesis. To illustrate the functional diversity of LU domain−containing proteins, we will discuss evolution, function, and medical relevance of selected members of this protein domain family.

## 2. Consensus Structures Defining LU Domains

Genes encoding LU domain proteins typically contain three exons: One for the N-terminal signal sequence followed by a set of two exons for the mature LU domain—generally flaked by phase-1 introns—thus facilitating genetic mobility (Figure 1A). Consistent with this composition, genes encoding LU-domain proteins often appear in small clusters in contiguous loci where they maintain their general intron−exon structure, as shown in Figure 1A. Such clustering of genes would suggest that an evolutionary expansion and diversification of this gene family occurred via multiple gene-duplication events. Accordingly, Loughner et al. [3] found that 30 out of the 48 LU containing-proteins in the human genome are located in just four small gene clusters. These segments are located on chromosomes 6p21 (*LY6G6C*, *LY6G6D*, *LY6G6F*, *LY6G5C*, and *LY6G5B*), 8q24 (*PSCA*, *LY6K*, *SLURP1*, *LYPD2*, *LYNX1/SLURP2*, *LY6D*, *GML*, *LY6E*, *LY6L*, *LY6H*, and *GPIHBP1*), 11q24.2 (*ACRV1, PATE1*, *PATE2*, *PATE3*, and *PATE4*), and 19q13 (*LYPD4*, *CD177*, *TEX101*, *LYPD3*, *PINLYP*, *PLAUR*, *LYPD5*, and *SPACA4*); the latter gene cluster includes all proteins in the human genome known to contain multiple LU domains [4]. The remaining LU domain encoding genes are more or less scattered in the human genome (i.e., *LYPD1*, *LYPD6*, *LYPD6B*, *LYPD8*, *CD59*, *BAMBI*, *ACVR1*, *ACVR1A*, *ACVR1B*, *ACVR1C*, *ACVR2A*, *ACVR2B*, *ACVRL1*, *BMPR1A*, *BMPR1B*, *BMPR2*, *TGFBR1*, and *TGFBR2*).

The consensus sequence defining the primordial LU domain comprises 60−90 residues with 10 plesiotypic cysteine residues engaged in a stereotypical disulfide-bonded network: 1–5, 2–3, 4–6, 7–8, and 9–10, as depicted by the sequence alignment of LU domains from different metazoan classes in Figure 1B. A non-glycosylated asparagine residue invariably follows the tenth cysteine in the LU domain signature. Notwithstanding the conservation of the LU domain signature, a high sequence diversity and a high propensity for undergoing lineage-specific expansion, diversification and neo-functionalization are the evolutionary hallmarks driving the functional versatility within this protein-domain family [5,6]. In some cases, this diversification and neo-functionalization even led to an erosion of the original plesiotypic disulfide pattern defining the LU domain, as illustrated in later sections. In particular, deletions of the 2–3 disulfide bond often occurred during this process (Section 3.1), but in very rare cases the 7–8 disulfide bond was also deleted (Section 4.3). Furthermore, additional apotypic disulfide bonds have occasionally been introduced into the LU domain scaffold.

Another salient feature of all LU domains is their unique protein-folding topology, where a cysteine-rich core projects three long β-hairpins (i.e., loops 1, 2, and 3) that assemble into a slightly curved central β-sheet, thus forming the dominating secondary structure of the characteristic three-fingered fold (Figure 1C). The six strands forming these three loops are designated A−F in the order of appearance in the primary sequence. These strands have a high propensity for forming β-sheets, with the exception of strand E, located at the edge of the LU domain, which can be flexible and adopt random coils, β-strands, or α-helices. The last disulfide bond (denoted 9–10) forms a small loop on the “back” of the central β-sheet, where it either terminates the LU domain in secreted proteins (Section 4.1) or extends into a carboxyl-terminal GPI-moiety that tethers the LU domain to the cell membrane in glycolipid-anchored variants (Section 4.2). The position of the intron, which divides the exon set encoding the mature LU domain, corresponds to the tip of loop 2 in the mature protein. The protruding loops and the concave face of the central β-sheet of the LU domains are generally involved in protein−protein interactions [7,8,9,10].

**Figure 1 ijms-20-02760-f001:**
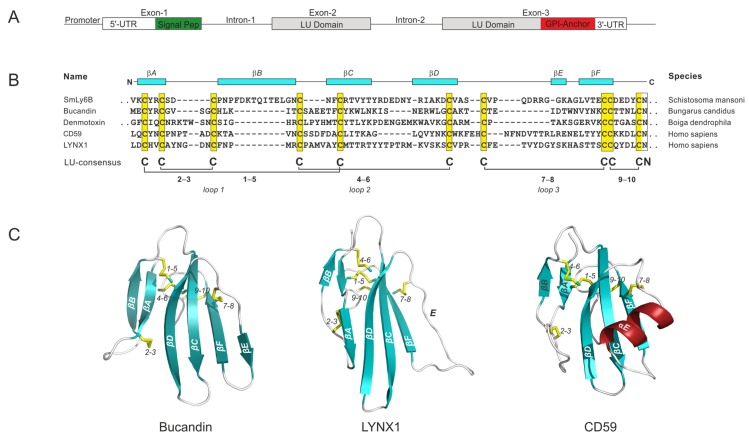
Characteristic properties of typical LU-domain encoding genes and their protein products: (**A**) The gene structure encoding an archetypical single LU-domain protein comprises three exons. Note that the signal sequence for GPI-anchoring is included in exon-3 for those proteins that are destined to become glycolipid anchored. (**B**) Sequence alignments of typical LU domains from different metazoan taxa where the plesiotypic 10-cysteine pattern is maintained (highlighted by yellow boxes along with the consensus disulfide bonding). Typical LU-domain secondary structure elements are shown in the top of the alignment as cyan boxes using the structure of Bucandin as reference. SmLy6B (Uniprot: B8Y6H3) from *Schistosoma mansoni* represents the class Trematoda [11]; Bucandin (Uniprot: P81782) from *Bungarus candidus* and Denmotoxin (Uniprot: Q06ZW0) from *Boiga dendrophilae* both represent the class Reptilia; CD59 (Uniprot: P13987) and LYNX1 (Uniprot: P0DP58) from *Homo sapiens* both represent the class Mammalia. Dots indicates an extension of the sequence. (**C**) The three dimensional protein structures of prototypical single LU-domain proteins are represented in a cartoon representation for CD59 [PDB 2OFS [12]], LYNX1 [PDB 2L03 [13]], and Bucandin [PDB 1F94 [14]]. The plesiotypic disulfide bonds are shown as yellow sticks and are numbered as in panel (**B**). The protruding strands forming the three loops are labelled *A*–*F*; β-sheets are colored cyan; α-helices are colored red.

## 3. Non-Mammalian LU-Domain Proteins

Although genes encoding LU-domain proteins are recognized in almost all phyla of the metazoan kingdom [15] and proteomics have revealed their presence in coelomic fluids of *Echiodermata* [16], we will only focus on a few examples of non-mammalian LU-domain proteins that have contributed significantly to our understanding of the evolutionary origin of the functional and structural diversity of LU domains.

### 3.1. Snake Venom α-Neurotoxins

Toxins from venomous snakes provide a rich source of information on the evolution of LU-domain containing proteins, in particular with a view to sequence diversification and neo-functionalization of α-neurotoxins. The co-evolutionary “arms race” between snake venom α-neurotoxins and their specific target proteins within the cholinergic system of their agile prey which they need to subdue provides a unique setting dominated by gene duplications and sequence evolution under positive Darwinian selection [6]. Extensive data mining of the numerous sequences from three-fingered toxins (more than 700 are known) has provided a unique insight into the rapid evolution and neo-functionalization of this scaffold. In this section, we will emphasize the diversification of the plesiotypic disulfide bonds in the snake toxins with respect to their specificity and efficacy in targeting essential receptors in their preferred prey.

Three-fingered toxins with the ancestral 10-cysteine LU-domain signature are the main constituent in venom from the advanced non-front fanged snake lineages (e.g., the genus *Boiga* in the family Colubridae). These toxins are often misclassified as “weak neurotoxins” due to their low toxicity towards synapsid targets (mammals). This is clearly a misnomer, since they are potent inhibitors of the cholinergic system of diapsids, which makes sense as these snakes feed primarily on birds, reptiles, and amphibians. Basal-type α-neurotoxin is, thus, a more appropriate terminology for these toxins, referring to their primordial phylogenetic origin. Within the framework for LU domains, an atypical covalently linked heterodimeric toxin, irditoxin [17], arose in the Colubridae family (Figure 2). From an evolutionary perspective, this represents an interesting case as the introduction of an eleventh cysteine into the LU domain occurred at different positions in the two subunits forming the heterodimeric irditoxins. It is likely that these changes occurred in concert, given that mutations introducing free cysteines in secreted proteins rarely survive selection because of the deleterious effects of the reactive free thiol group [18]. Irditoxin possesses a high taxon-specific lethality, since its blockage of avian neuromuscular junctions is 1000-fold more potent than blockage of the corresponding neuromuscular junctions in mammals [17]. The evolution of irditoxin—a toxin that is more potent than the single LU-domain toxin denmotoxin—is probably among the driving factors for the “success” of *Boiga irregularis* as an invasive species in the Pacific island of Guam [17,19].

An impressive radiation in toxin diversification and potency towards synapsids arose in the advanced snake lineage Elapidae subsequent to the anatomical acquisition of a high-pressured and hollow front-fanged venom-delivery system. Evolution of this delivery system was tightly associated with the neofunctionalization of three-fingered toxins. This occurred primarily via the selective deletion of one plesiotypic LU-domain disulfide bond—the one that stabilizes loop 1 and is denoted 2–3 in Figure 1 and Figure 2. One hypothesis proposes that the loss of the structural constraints from this disulfide bond created a more flexible toxin scaffold, which subsequently facilitated neo-functionalization by rapid diversification of surface exposed residues [6]. The resultant 8-cysteine LU-domain scaffold contributed to high potency towards many mammalian targets, resulting in the notorious toxicity of elapid snake venom in humans. Short-chain α-neurotoxins gained high potency towards mammalian nicotinic acetylcholine receptors (α1 nAChR), breaching the taxon-specific lethality for the toxins found in colubrine snakes with the complete 10-cysteine LU-domain signature. The introduction of an apotypic disulfide bond at the tip of loop 2 in the LU domain of the long-chain α-neurotoxins (Figure 2) further expanded their targeting repertoire to include α7 nAChR. A subgroup of the long-chain α-neurotoxins developed into non-covalent homodimeric toxins (e.g., κ-bungarotoxin), which antagonizes the neuronal α3β2 nAChR. Along the same lines, haditoxin [20], which is a homodimeric short-chain α-neurotoxin, also exhibits a broad pharmacologic specificity by targeting muscle as well as several neuronal nAChRs (α7, α3β2, α4β2). The high adaptability of the 8-cysteine LU-domain scaffold for undergoing neo-functionalization is clearly illustrated by the wide range of targets that it can antagonize. Besides nicotinic acetylcholine receptor antagonists, these toxins can act as muscarinic acetylcholine receptor antagonists (MT7), acetylcholinesterase inhibitors (fasiculins), L-type calcium channel antagonists (calsiceptine), non-specific cytotoxins disrupting the phospholipid bilayer (cardiotoxins), or as modulators of the acid-sensing ion channels (mambalgins). Intriguingly, mambalgins exhibit no toxic effects, but by inhibiting the acid-sensing ion channels, they exhibit potent analgesic effects (comparable to morphine) without inducing tolerance or respiratory distress. This profile triggered considerable pharmacological interest in these LU-domain proteins as therapeutic agents to alleviate chronic pain [9,21].

**Figure 2 ijms-20-02760-f002:**
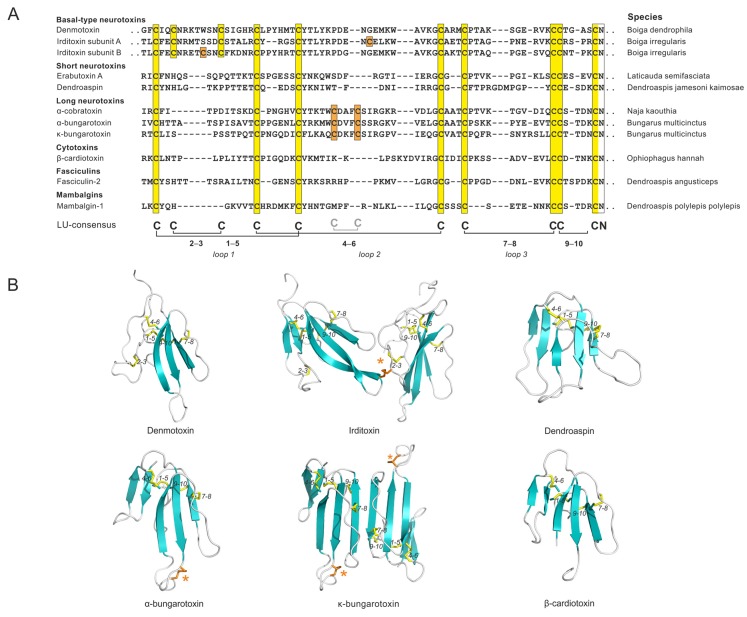
Evolution and neo-functionalization of LU domains from snake venom toxins: (**A**) A sequence alignment of typical members of the different groups of snake-venom toxins. Note, only the basal-type neurotoxins maintain the 2–3 disulfide bond. Orange boxes highlight apotypic cysteine residues and yellow boxes highlight plesiotypic LU-domain cysteine residues. Dots indicate sequence extensions. The apotypic, intra-domain disulfide bond in the long-chain α-neurotoxins is included in the consensus sequence in light gray. (**B**) Three-dimensional structures of selected LU domains belonging to basal-type α-neurotoxins [denmotoxin (PDB 2H5F [22]); irditoxin (PDB 2H7Z [17])], short-chain α-neurotoxins [dendroaspin (PDB 2LA1 [23])], long-chain α-neurotoxins [α-bungarotoxin (PDB 1HC9 [24]); κ-bungarotoxin (PDB 1KBA [25])], and cytotoxins [β-cardiotoxin (PDB 3PLC)]. The apotypic disulfide bonds in the long-chain α-neurotoxins and irditoxin are marked with an asterisk.

Of note, a dynamic recruitment of genes to the postorbital venom gland appears to have evolved by distinct co-option events of genes expressed in other tissues that are supporting normal physiological processes [26,27,28]. That concept—where LU domains with normal and toxic functions could have a shared phylogenetic ancestry—is consistent with the observation that LYNX1 (Figure 1) and SLURP1 (Section 4.1), both prototypical LU domains with 10 cysteine residues, have modulatory roles on nAChR activities in the normal brain [29,30] and skin [31], respectively. Such an evolutionary trajectory would be in accordance with the species-selectivity of the early snake-venom toxins for diapsid targets. In line with this theory for neurotoxin evolution, LU-domain proteins, involved in regulating the activity of acetylcholine receptors, are often referred to as endogenous prototoxins. One study demonstrated that expression of α-bungarotoxin in zebrafish muscle fibers in vivo in a tethered version by adding a GPI-anchor specifically silenced muscle nAChR activity without having systemic toxicity or overt effects of neuromuscular synapse development, thus reverting this toxin into a “prototoxin-like” state [32].

### 3.2. LU Domain Proteins in Drospohila

LU domains are also widely expressed within the class Insecta, where their expression and function in Drosophilae has been studied intensively. The genome of *Drosophila* contains 45 genes encoding LU-domain proteins, all of which are predicted to be GPI-anchored [33]. As observed in other metazoan classes, Drosophilae genes encoding LU-domain proteins tend to cluster in specific regions of the genome, suggesting that they have arisen by multiple gene-duplication events and subsequent diversification [5,34]. More than half of these genes encode archetypical single LU-domain proteins with 10 cysteines and a C-terminal signal sequence for GPI-anchoring [33]. A few proteins contain multiple LU domains (ranging from 2 to 44 LU domains). Among these proteins, two contain LU domains with only 8 disulfide bonds (lacking the 7–8 plesiotypic disulfide bond), but their expression in vivo has not yet been verified [33].

From a functional perspective, four *Drosophila* genes encoding GPI-anchored proteins with a single LU domain (*boudin*, *crooked*, *coiled*, *crimpled*) have attracted considerable interest. These single LU-domain proteins play non-redundant roles in establishing the epithelial septate junctions in Drosophilae, which function as anatomical diffusion barriers equivalent to tight junctions in vertebrate epithelia [33,35,36]. The precise mode of action of these LU-domain proteins is unclear, but it may involve the trafficking of septate junction constituents such as Neurexin IV [35,36].

Interestingly, the *Drosophila* brain contains another GPI-anchored, single LU-domain protein resembling an endogenous prototoxin. This LU-domain protein, encoded by *quiver/sleepless*, possesses a remarkable dual functionality. It both antagonizes the nAChR encoded by *redeye* [37] and stimulates the voltage-gated potassium channel *shaker* [38]. These properties endow quiver/sleepless with the ability to modulate neuronal excitability and cholinergic synaptic transmission, serving to regulate sleeping patterns in *Drosophila* [39]. This observation suggests that the interaction between endogenous prototoxins and nAChRs is a more general phenomenon, present in metazoan classes as diverse as Mammalia (*LYNX1*) and Insecta (*sleepless*).

### 3.3. LU-Domain Proteins in Teleosts

From the standpoint of evolution, the wholesale genome duplication that occurred at the base of the teleost radiation provides another interesting case involving diversification of two paralogous genes. The fate of duplicated genes may entail neo-functionalization, functional conservation, or drift into a silenced pseudogene [40]. Although the study on teleost LU-domain proteins is in its infancy, it is clear that genomes of zebrafish and medaka contain small contiguous clusters of genes with the prototypical intron−exon structure of plesiotypic LU-domain proteins [41,42,43,44,45]. A cluster on zebrafish chromosome 21 encodes seven GPI-anchored proteins containing two consecutive LU domains [42]; six of these proteins are expressed primarily in the developing brain and one is expressed in the skin. Both LU domains in these proteins maintain the 10-cysteine signature. In contrast, all mammalian proteins containing multiple LU domains lack the 7–8 plesiotypic disulfide bond in the N-terminal LU domain (Section 4.3).

Herberg et al. [41] demonstrated that one GPI-anchored, single LU-domain protein, *bouncer*, is expressed on zebrafish oocytes and is required for fertilization, as it mediates the contact between the oocyte and the spermatocyte. The closest human homolog of *bouncer* is *SPACA4/SAMP14* (sperm acrosomal membrane protein 14). Herberg et al. proposed that *bouncer* is one of the key components governing species-specific fertilization in teleosts. This conclusion is based mainly on cross-fertilization experiments after swapping the endogenous expression of *bouncer* in zebrafish oocytes with that of medaka. This swapping strategy allowed the entry of medaka sperm into zebrafish oocytes, albeit at a low efficiency. As illustrated in Figure 3, the gene encoding *bouncer* occurs in two paralogous forms (A and B) in medaka, due to the early gene duplication. One variant contains the 10-cysteine LU-domain signature (B), whereas the other (A) has lost the 2–3 plesiotypic disulfide bond that in snakes induced rapid diversification and neo-functionalization of the α-neurotoxins (Section 3.1). Replacing the 10-cysteine LU-domain variant of zebrafish *bouncer* with the 8-cysteine LU-domain variant from medaka (A) may therefore have lowered the efficacy by which cross-fertilization occurred. It is possible that the B-form of medaka *bouncer* does not produce a correctly folded protein, as it contains an unpaired eleventh cysteine residue (Figure 3). As the A-variant of *bouncer* from *Cyprinus carpio* maintains a similar 10-cysteine LU-domain signature as that found in zebrafish *bouncer*, it would be interesting to test if the swapping protocol used by Herberg et al. would lead to a higher cross-fertilization efficacy between these species.

The importance of another gene (*lypd6;* LY6/PLAUR domain-containing 6) encoding a GPI-anchored LU-domain protein on early zebrafish development is also clearly documented [45]. By genetically manipulating *lypd6*, it was shown that this protein regulates embryonic mesoderm and neuroectoderm patterning by enhancing Wnt/β-catenin signaling via binding to Lrp6 in lipid rafts [45]. *Lypd6* contains an additional apotypic disulfide bond stabilizing its third loop (as illustrated by the structure of human LYPD6 in Figure 6D).

## 4. Mammalian LU Domain Proteins

### 4.1. Secreted Single LU Domain Proteins

Of the 48 genes encoding LU-domain proteins in the human genome, 11 encode a secreted version of a single LU-domain protein, such as SP-10, PATE 1–4, SLURP-1, and SLURP-2 (secreted Ly6/uPAR-related proteins). These secreted proteins retain the genetic and structural hallmarks of LU domains. Among the secreted LU-domain proteins, SLURP-1 has received the most attention, since missense mutations in that gene cause a rare autosomal-recessive skin disease, *mal de Meleda* [46]. Patients with *mal de Meleda* exhibit palmoplantar keratoderma with transgrediens. SLURP-1 is expressed primarily in the *stratum granulosum* of the epidermis [47]. Several of the missense mutations in SLURP-1 associated with *mal de Meleda* affect one of the 10 plesiotypic LU-domain cysteine residues (pCys77Arg, pCys94Ser, and pCys99Tyr [46]). These mutations grossly impair the folding of the LU domain, preventing efficient secretion from cells [48]. Deletions of either SLURP-1 or SLURP-2 leads to a *mal de Meleda*-like phenotype in mice and the combined double deficiency causes a comparable disease severity, as presented by the individual single deficiencies, suggesting that SLURP-1 and SLURP-2 either act together or act sequentially in the same pathway [49,50,51]. SLURP-1 inhibits keratinocyte proliferation in vitro by 40%, presumably by antagonizing binding to the α7-nAChR with low nanomolar affinities [31], while SLURP-2 in contrast stimulates keratinocyte proliferation in vitro and presents a more promiscuous binding profile towards several AChRs [52]. Whether these effects are causally related to development of *mal de Meleda* remains unclear.

### 4.2. Glycolipid-Anchored Single LU-Domain Proteins

The majority of LU-domain proteins encoded in the human genome are GPI-anchored single-domain proteins with the 10-cysteine signature. Although protein structures and biological functions of a few of these proteins are well-characterized (e.g., CD59, GPIHBP1, LYNX1, LYPD6), molecular and functional insights into the majority of these family members are at best rudimentary.

#### 4.2.1. CD59

One of the best-characterized proteins within this group is the complement regulatory protein CD59 (Figure 1). CD59 protects host cells from autologous complement damage by binding to the premature membrane attack complex C5b–8, thus preventing maturation into the terminal pore-forming cytolytic complex. Phylogenetically, CD59 exhibits a broad taxonomic distribution in vertebrates, spanning from teleost to mammals, but CD59 is lacking in *Cavia porcellus* (guinea pig), where the CD59 gene has been transformed into a pseudogene [53]. A few rare cases of homozygous missense mutations leading to defective CD59 have been identified in humans [54,55,56,57]. These defects are associated with a life-threating prothrombotic phenotype with intravascular hemolysis, cerebral infarction, and relapsing peripheral neuropathy. The ability of rodents to withstand CD59 deficiency could be due to the protective activity of another complement regulatory component (Crry) in those species [58]. One of the two deleterious single-site missense mutations in human CD59 disrupts the 9–10 disulfide bond (pCys64Tyr). This mutation destabilizes CD59 folding and interferes with transport of the protein to cell surface [57], thus providing the molecular basis for its association with disease development.

#### 4.2.2. LYNX1

Studies on mice with genetic ablation of *Lynx1* reveal that this prototoxin limits neuronal plasticity in the adult visual cortex by attenuating the cholinergic response of α4β2 and α7 nAChRs [29,59]. *Lynx1* is widely expressed in a variety of neuronal subtypes in the brain where it colocalizes with α4β2 and α7 nAChRs [32,60]. The progressive increase in *Lynx1* expression in the visual cortex neurons of the developing brain thus gradually impair visual acuity after monocular deprivation (amblyopia) in adults versus juveniles, but importantly this limitation of adult mice is rescued by increased neuronal plasticity in *Lynx1* deficient mice [29]. Pharmacological intervention via administration of an acetylcholinesterase inhibitor (physostigmine) also induces neuronal plasticity in the adult mouse brain [29]. The impact of *Lynx1* on the complex regulation of cholinergic output is nonetheless not restricted to the visual cortex, but includes additional functions, such as motor learning and associative learning [30,61]. The integrity of the 2–3 plesiotypic disulfide bond in the LU domain of Lynx1 is essential for its nAChR modulating function [60,62], which is in contrast to observations with CD59, uPAR DI, and κ-bungarotoxin where this particular disulfide bond is non-essential for the function of these proteins. One study reports that another GPI-anchored LU-domain protein, LYPD6, also interacts with and modulates nAChR function [63]. A more comprehensive review on the functional aspects of endogenous LU domain modulators of nAChRs is found elsewhere [64].

#### 4.2.3. GPIHBP1

From an evolutionary perspective, the inclusion of GPIHBP1 in the LU-domain protein superfamily represents a recent event, as this protein occurs exclusively in the class Mammalia [65]. GPIHBP1 serves an important role in delivering lipids to oxidative tissues such as heart and muscles by focusing active triglyceride hydrolysis to the lumen of capillaries [66]. Several of the essential steps in this complex process are regulated by GPIHBP1: (i) Shuttling of the lipoprotein lipase from the interstitial spaces (where it is secreted by parenchymal cells) to the capillary lumen is exclusively dependent on GPIHBP1 [67]; (ii) margination of triglyceride-rich chylomicrons on the endothelial membrane is mediated by the GPIHBP1•LPL complex [68]; (iii) extraction of LPL from a dynamic pool, loosely tethered to heparan sulfate proteoglycans, is driven by GPIHBP1 [69]; (iv) stabilization of LPL structure and activity is accomplished by GPIHBP1 binding [70]; and (v) protection from the endogenous protein inhibitors ANGPTL4 and ANGPTL3/8 is also accomplished by GPIHBP1 binding [71]. To perform these roles, GPIHBP1 developed a number of unique properties, which partly were made possible by the addition of an extra exon in front of the exon-set encoding the generic GPI-anchored LU domain (Figure 4A). Remarkably, this exon encodes a highly acidic N-terminal extension with 21 negatively charged residues (Glu or Asp) as well as a sulfated tyrosine [69] within 26 consecutive residues in human GPIHBP1 (Figure 4B). The length of this extension is highly variable among mammalian species and can be as long as 50 amino acid residues, including 32 negative charges (*Monodelfis domestica*; XP_016287565.1). The evolutionary origin of the additional exon-2 in the GPIHBP1 gene remains unclear, but it was speculated to have arisen from integration of a segment of the *BCL11A* gene [72].

The acidic extension, which is intrinsically disordered, endows GPIHBP1 with several unique functional properties. First, it dramatically increases the encounter rate with LPL due to electrostatic steering; the association rate constant (*k_on_*) between LPL and GPIHBP1 is thus >250-fold greater for full-length GPIHBP1 than for a mutant lacking the acidic N-terminal extension [69]. Second, GPIHBP1′s acidic N-terminal extension is crucial for the ability of GPIHBP1 to extract LPL from heparan sulfate proteoglycans in the subendothelial space [69]. Third, GPIHBP1′s intrinsically disordered extension has a chaperon-like function, blocking the tendency of LPL to unfold [70]. Finally, GPIHBP1 limits the unfolding of LPL catalyzed by its physiologic inhibitors, the ANGPTL proteins [71]. The entire concave face of the central β-sheet and the three protruding loops of GPIHBP1′s LU-domain participate in a hydrophobic binding interface with LPL [7], adding stability to the LPL•GPIHBP1 complex (Figure 4C).

Any defect in the assembly of the LPL•GPIHBP1 complex causes severe hypertriglyceridemia (chylomicronemia)—a condition associated with life-threatening bouts of acute pancreatitis. Chylomicronemia is lifelong in the setting of homozygosity or compound heterozygosity for loss-of-function mutations in GPIHBP1 or LPL [74,75]. Several of these disease-causing missense mutations in human GPIHBP1 involve elimination of one of the plesiotypic cysteine residues in the LU domain (e.g., pCys65Tyr, pCys65Ser, pCys68Tyr, pCys68Gly, pCys83Arg, pCys89Phe), leaving the partner half-cystine with an unpaired thiol-group [74]. In one case, the deleterious mutation actually introduced a new unpaired cysteine in the LU domain of GPIHBP1 (pSer107Cys) [76]. The severe phenotypes of these patients are most likely caused by the destabilizing of the LU-fold leading to multimerization of dysfunctional mutant protein [77].

Acquired forms of chylomicronemia can occasionally occur in children or adults as a result of autoantibodies against GPIHBP1 [78,79]. These autoantibodies, which are directed against the LU domain of GPIHBP1, abolish the ability of GPIHBP1 to bind LPL. Consequently, LPL cannot reach its site of action in the capillary lumen. Approximately one-half of patients with GPIHBP1 autoantibodies have clinical or serologic evidence for autoimmune diseases.

#### 4.2.4. LY6E

Two interferon inducible LU genes (*LY6E* and *PSCA*) have adverse pathogenic effects, as they enhance the susceptibility of certain cell types to a subset of viral infections. Host entry of Flaviviridae, such as Zika virus, dengue virus, and yellow fever virus, occurs via clathrin-mediated endocytosis, but the size of these virion particles requires the active engagement of a non-canonical endocytosis pathway, which includes the GPI-anchored LU domain protein LY6E [80]. A different mechanism for enhanced viral infection revealed that influenza A rely on *LY6E* for promoting disassembly of the viral capsid (uncoating) after endosomal escape of the internalized virus. How LY6E aids disassembly of the capsid proteins remains nevertheless unclear, but the base of loop 1 in the LU domain of LY6E seems to play an essential role in this process. Possible mechanistic insights into the LY6E-facilitated entry of viruses may perhaps be gleaned upon from studies on the biological function of LY6E in normal physiology. *Ly6e*-deficient mice show mid-gestational embryonic lethality (E15.5) due to placental malfunction with impaired labyrinth morphogenesis and imperfect syncytiotrophoblast fusion [81]. This phenotype relates to Ly6e being the endogenous receptor for syncytiotrophoblast layer fusogenic protein A (Syncytin A), which is encoded by *Syna*, an ancient retroviral envelope gene that was co-opted in Mammalia to mediate fusion of distinct placental cells into functional syncytiotrophoblasts [82].

### 4.3. Glycolipid-Anchored Proteins with Multiple LU Domains

The human genome contains a small locus on chromosome 19q13 that encodes atypical LU domain-containing proteins (*LYPD4, CD177, TEX101, LYPD3, PINLYP, PLAUR, LYPD5, SPACA4*). Several of these genes encode GPI-anchored proteins with two or more LU domains with the generic intron−exon structure preserved for each added LU domain. As a completely unexpected and unique feature, the N-terminal LU domain in all these multi-LU-domain proteins lack the 7−8 plesiotypic disulfide bond [4,83]. Deleting that particular disulfide bond in the single LU-domain proteins invariably leads to an unstable and aggregated recombinant protein product, implying that this disulfide bond is essential for integrity of a proper folded LU domain [77,84,85]. In this section, we will focus on three GPI-anchored proteins from this locus: The urokinase-type plasminogen activator receptor uPAR (*PLAUR*) with three consecutive LU domains and the two LU domain-containing proteins, C4.4A (*LYPD3*) and Haldisin (*LYDP5*).

The best-characterized member of these glycolipid-anchored, multi-LU-domain proteins is the urokinase-type plasminogen activator (uPA) receptor (uPAR), which is also the founding member of the LU domain superfamily. In a functional context, uPAR serves to focus uPA-mediated plasminogen activation on the cell surface though high-affinity interaction with the growth factor-like domain of uPA (Figure 5D). One important function of this cell-surface plasminogen activation system is to provide a “clean-up” mechanism for extravascular fibrin. With aging, mice deficient in uPAR show signs of chronic hepatic inflammation due to accumulating fibrin deposition [86], and they also have an impaired neuronal recovery after cerebral ischemia [87,88]. Notwithstanding the beneficial function of uPAR, high expression levels of uPAR and uPA may also elicit detrimental pathological effects, particularly in the setting of chronic inflammation. Progression of arthritic lesions seems to be exacerbated by the presence of a high expression levels of uPA and uPAR [89,90]. Likewise, numerous studies have demonstrated that high levels of uPAR predict poor survival for patients with solid cancers [91]. These observations have prompted several strategies for uPAR-targeted treatment [92,93,94,95]. In addition, they have triggered the development of non-invasive PET-imaging modalities designed to visualize uPAR expression in cancer patients by PET-imaging, with the goal of improved patient stratification [96,97,98]. Optical imaging of uPAR expression with near-infrared fluorescence is also currently being pursued as an intra-operative tool in guiding precision cancer surgery [99,100].

The two key physiological binding partners for uPAR, uPA and vitronectin, bind uPAR with markedly different affinities (K_D_’s for uPA and vitronectin are 0.02 nM and 4 µM, respectively). A dynamic assembly of all three LU domains in uPAR creates a large hydrophobic uPA-binding cavity involving the concave faces of all of the central β-sheets of its LU domains [101,102,103,104,105,106]. Biophysical studies have demonstrated that uPAR DI (the first LU domain) is highly flexible and exhibits a dynamic association with DII and DIII, but this inter-domain interface is far more rigid after uPA-binding [106]. This relationship is remarkable, given that uPAR DI lacks the plesiotypic 7–8 disulfide bond, which is indispensable for the folding of single LU-domain proteins. Moreover, this particular disulfide bond stabilizes loop 3 of the LU domain, which is engaged in the interface between uPAR’s first and second LU domain (Figure 5C). We therefore propose that some flexibility of this scaffold is needed for the assembly of the LU domains in intact, unoccupied uPAR. Supporting this assumption, we showed that reintroducing the 7–8 disulfide bond into the first LU domain of uPAR impairs uPA binding as well as the dynamic association between DI and DII-DIII in the unoccupied receptor [107]. From an evolutionary perspective, it is noteworthy that all uPAR orthologues identified thus far in Mammalia and Reptilia have three consecutive LU domains, and in each case the N-terminal LU domain lacks the plesiotypic 7–8 disulfide bond [107,108]. The uPAR-like proteins with three consecutive LU domains identified in Sarcopterygii and Amphibia maintain a generic 10–cysteine pattern in each of the three LU domains [108]. However, the uPAR-binding sequences in uPA (as defined within Mammalia) are only present in those species where the 7–8 disulfide bond in the first LU domain of uPAR is absent [107].

Another pair of genes, *LYPD3* and *LYPD5*, located in the same locus as uPAR on chromosome 19q13, encode two GPI-anchored proteins, which are robust biomarkers of epithelial differentiation. C4.4A/*LYPD3* is confined to *stratum spinosum* [109,110,111], and Haldisin/*LYPD5* is confined to *stratum granulosum* [112]. Both proteins contain two LU domains and the aforementioned 7–8 disulfide bond is absent from their N-terminal LU domain (Figure 5A). In addition, the first LU domain of Haldisin lacks the 2–3 disulfide bond, resulting in a LU domain containing only three of the five plesiotypic disulfide bonds. The biological function of these proteins in the stratified squamous epithelium is unclear, and mice deficient in C4.4A manifest only minor phenotypes [113]. Nonetheless, several independent studies have shown that high levels of C4.4A expression in pulmonary non-small cell adenocarcinomas predicts poor patient survival [114,115,116].

### 4.4. Transmembrane Proteins with a Single Extracellular LU Domain

It is possible that the LU domain, in an evolutionary context, first appeared as an extracellular ligand-binding domain in the primordial TGF–β signaling receptors. These receptors are essential for embryogenesis and ontogenesis of multicellular organisms, and they are already present in primitive bilaterian metazoans with elaborate body plans [15,117]. This important class of signaling molecules comprises a large group of agonists, antagonists, anchoring molecules (e.g., latent TGF-β binding protein), signaling receptors (type I and type II), and co-receptors [117]. The co-evolution, protein structures, and molecular mechanisms defining this system have been thoroughly investigated. A more detailed description can be found in a comprehensive and contemporary review by Hinck et al. [117]. A central event in this signaling pathway is driven by the heterodimerization of two integral membrane receptors by ligand binding to their extracellular domains (ECD). The ECD of type I receptors (e.g., TGF-βR1, BMPR1A, and ACVR1A) all comply with the plesiotypic LU domain signature with 10 cysteines and the stereotypic disulfide bonding pattern (Figure 6A). In contrast, ECDs of type II receptors have a more divergent cysteine pattern and a longer loop 1. In BMPR2 and ACVR2, the ECD has lost the 2–3 disulfide bond and gained another apotypic disulfide bond tethering strand E to the back of the three-fingered scaffold (Figure 6C). This cysteine configuration resembles the one found in LYPD6, where an apotypic disulfide bond also stabilizes loop 3, albeit at a more distal position (Figure 6D). The ECD of TGF-βR2 represents the most divergent member of this family. This domain has lost the 7–8 plesiotypic disulfide bond, but gained two additional apotypic disulfide bonds stabilizing loop 1 and loop 3 (Figure 6).

## 5. Conclusions

The LU domain is widespread in the Metazoa kingdom, where it carries out an extremely diverse set of biological functions. Although this domain is encoded by an exon-set with symmetrical intron−exon boundaries (mostly of phase 1), it probably cannot be considered a bona fide mobile protein module, as it is found predominantly as single LU-domain proteins or as repetitive units in multidomain proteins containing only this domain. However, it would be entirely reasonable to propose the LU domain as a “proto-module”, given that it is found in the context of a non-homologous protein environment in a few proteins, for example, GPIHBP1, SP-10 and the ECD of TGF-β receptors.

## Figures and Tables

**Figure 3 ijms-20-02760-f003:**
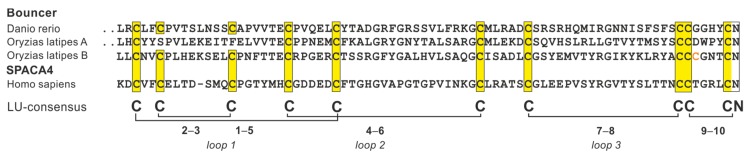
Sequence alignment of *bouncer* from zebra fish (*Danio rerio*) with two paralogous sequences (**A** and **B**) from medaka (*Oryzias lapites*). These proteins are expressed by oocytes. The B variant of medaka bouncer contains an unpaired cysteine (highlighted in orange) and lacks a functional C-terminal signal sequence entailing membrane tethering by a GPI-anchor. Also shown is the sequence from the closest human homolog, SPACA4/SAMP14, expressed in spermatocytes.

**Figure 4 ijms-20-02760-f004:**
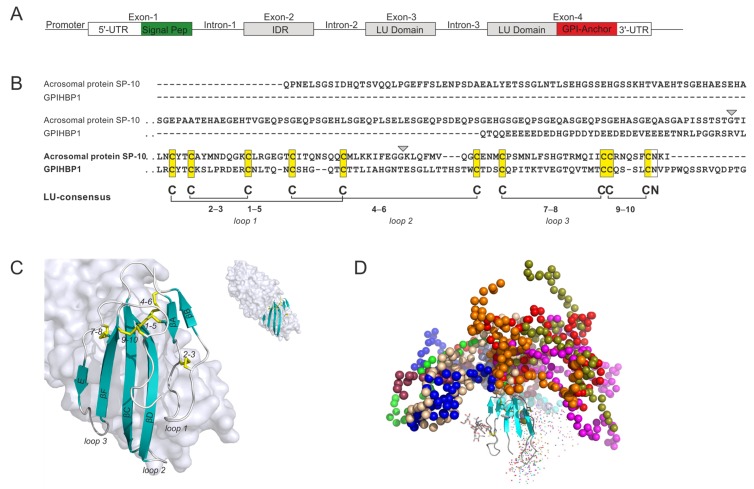
Gene structure, sequence, and three-dimensional structure of GPIHBP1: (**A**) The gene organization of GPIHBP1, containing an additional exon (*IDR*) in front of the traditional exon-set encoding the LU domain. (**B**) A sequence alignment of the LU domains of GPIHBP1 and SP-10, the only two proteins in the human genome with an extra exon encoding an intrinsically disordered N-terminal segment. Gray arrowheads highlight the positions of introns 2 and 3. The length of the IDR extension of SP-10 varies considerably due to alternative splicing events in exon 2 [73]. (**C**) The crystal structure of GPIHBP1 bound to the lipoprotein lipase (LPL). The gray surface represents LPL, whereas the LU domain of GPIHBP1 is shown as a cartoon representation, using the same color-coding as in the earlier figures [PDB 6E7K [7]]. Only the LU domain is defined in the crystal structure; as the acidic intrinsically disordered domain at the amino terminus is not well defined in the electron density map and most likely forms a fuzzy complex with LPL. (**D**) A model of GPIHBP1 based on small-angle X-ray scattering, with the likely spatial distribution of the acidic disordered extension illustrated with colored beads, each color representing one likely spatial distribution. Reproduced with permission from Kristensen et al. [69].

**Figure 5 ijms-20-02760-f005:**
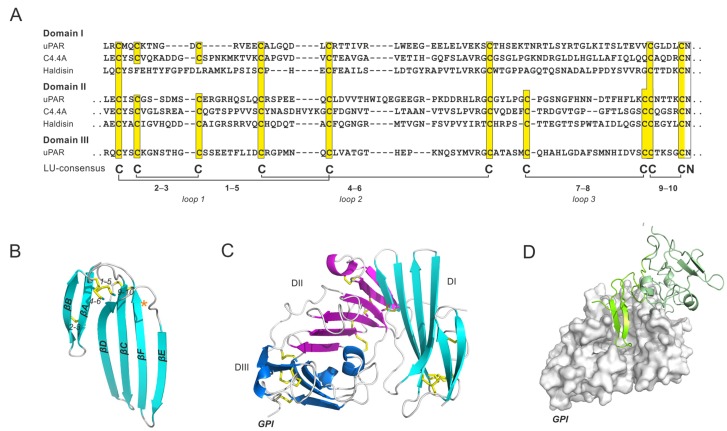
Multi-LU-domain proteins. (**A**) A sequence alignment of three human multi-LU-domain proteins: uPAR/*PLAUR* (Uniprot: Q03405), C4.4A/*LYPD3* (Uniprot: O95274) and Haldisin/*LYPD5* (Unitprot: Q6UWN5). This alignment includes the first two LU domains in uPAR, C4.4A and Haldisin, along with third LU domain in uPAR. Note, that the plesiotypic 7−8 disulfide bond is lacking in all the N-terminal LU domains. (**B**) Cartoon representation showing that the structure of the N-terminal LU domain of uPAR with the position of the missing 7−8 disulfide highlighted by an asterisk. Disulfide bonds are shown as yellow sticks. (**C**) Cartoon representation illustrating the assembly of the three LU domains in intact uPAR, with DI in cyan, DII in purple, and DIII in blue. The position of the glycolipid-anchor that tethers uPAR to the cell membrane is shown (GPI). (**D**) The complex between uPAR (gray surface representation) and the amino-terminal fragment (ATF) of its primary high-affinity ligand uPA (shown in a green cartoon representation). The structures were created by PyMol with the PDB coordinates 3BT1 [101].

**Figure 6 ijms-20-02760-f006:**
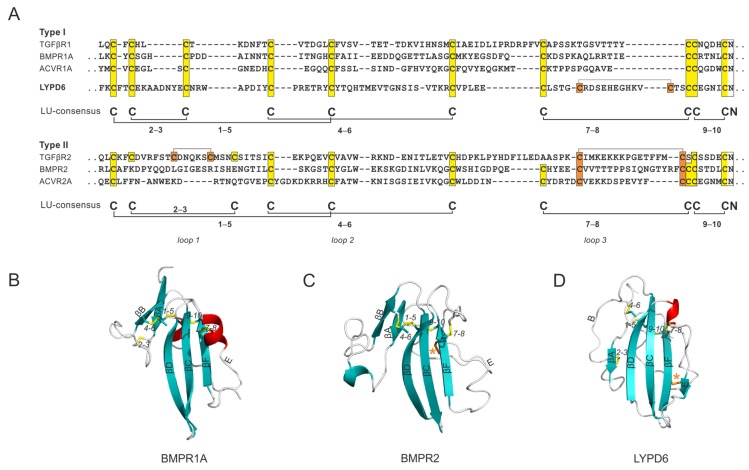
Integral membrane receptors in which the LU domains function as extracellular ligand-binding domains. (**A**) A sequence alignment of the extracellular N-terminal ligand-binding LU domain of the TGF–β receptor 1 (Uniprot: P36897), bone morphogenetic protein receptor 1A (Uniprot: P36894), activing receptor 1A (Uniprot: Q04771), LYPD6 (Uniprot: Q86478), TGF–β receptor 2 (Uniprot: P37173), bone morphogenetic protein receptor 2 (Uniprot: Q13873), and activin receptor 2 (Uniprot: P27037). The separation of the alignment for type 1 and type 2 receptors emphasizes the preservation of the LU signature in type 1 receptors and the introduction of apotypic disulfide bonds in the type 2 receptors (highlighted by orange boxes). Representative structures of (**B**) bone morphogenetic protein receptor 1A [PDB: 1REW [118]]; (**C**) bone morphogenetic protein receptor 2 [PDB: 2HLQ [119]]; and (**D**) LYPD6 (PDB: 6GBI [120]). Orange asterisks mark positions of the apotypic disulfide bonds.

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
