# Peer review of "Evolution and Medical Significance of LU Domain−Containing Proteins"

_ijms, 2019, doi:10.3390/ijms20112760_

Round 1
Reviewer 1 Report
Now it is ready to publish
Author Response
Thank you for your comments. We appreciate your efforts on our manuscript.
Reviewer 2 Report
The manuscript of Leth et al was greatly improved. At present the manuscript is written in clear manner, and reading of it is very interesting and educational. Nevertheless, I have several minor remarks.
Lane 193. Please, remove a7, because a7-nAChRs are neuronal receptors and do not express in muscle fibers.
I did not understand why the authors discuss the Lypd6 protein in the section 4.4, although they simultaneously mention this protein in the section 4.2 (I believe that it is better place for it). I think Lypd6 should be moved from the chapter 4.4 to the chapter 4.2. Lypd6 is not a receptor, it is a ligand of nAChRs and LRP6 (Arvaniti et, 2016; Zhao et al, 2018).
Recently it was shown that Lynx1 and SLURP-2 are not the products of one gene (Loughner et al, 2016), it is incorrect to write that these proteins are homologs (line 305).
Line 308. The reference is incorrect. Citation of Ines Ibanez-Tallon et al, Neuron, 2002 should be there.
Recently, LU-domain proteins were found also in starfish Asterias rubens (Shabelnikov et al, 2019). Please discuss shortly this work.
Author Response
The manuscript of Leth et al was greatly improved. At present the manuscript is written in clear manner, and reading of it is very interesting and educational. Nevertheless, I have several minor remarks.
Lane 193. Please, remove a7, because a7-nAChRs are neuronal receptors and do not express in muscle fibers. Corrected, thanks for pointing this out.
I did not understand why the authors discuss the Lypd6 protein in the section 4.4, although they
simultaneously mention this protein in the section 4.2 (I believe that it is better place for it). I think Lypd6 should be moved from the chapter 4.4 to the chapter 4.2. Lypd6 is not a receptor, it is a ligand of nAChRs and LRP6 (Arvaniti et, 2016; Zhao et al, 2018). We did actually put the functional discussion of LYPD6 in section 3.3 focusing on Teleosts as the importance of this protein for development was unraveled in zebrafish. The only reason that the sequence appears in Figure 6 is for structural comparison, as it most resembles the organization of TGFβ-RI and TGFβ-RII – as emphasized by Zhao et al, 2018. To emphasize a possible role in regulation of nAChR function we have now added this information to section 4.2.2.
Recently it was shown that Lynx1 and SLURP-2 are not the products of one gene (Loughner et al, 2016), it is incorrect to write that these proteins are homologs (line 305). This has now been removed.
Line 308. The reference is incorrect. Citation of Ines Ibanez-Tallon et al, Neuron, 2002 should be there. That reference has now been added.
Recently, LU-domain proteins were found also in starfish Asterias rubens (Shabelnikov et al, 2019). Please discuss shortly this work. We have now added this reference to the information stated at lines 114-115, but as there is no hardcore structure-function data presented in this study, we did not consider it relevant to discuss the proteomic data in more detail.
Reviewer 3 Report
This revised version addresses all major concerns I expressed. The text has been significantly improved and I recommend publication of this review in its present form.
Author Response
Thanks for your comments. We really appreciate your efforts on our manuscript.
This manuscript is a resubmission of an earlier submission. The following is a list of the peer review reports and author responses from that submission.
Round 1
Reviewer 1 Report
The abstract and conclusions should reflect better for the whole paper. When taking published data, making own figures without corresponding paper citation is not proper. Figure 4D is essentially same as in Ref 46: Fig6G, this may cause a copyright problem.
Very relevant point. We published the PNAS paper in question (Ref 46) with the copyrights licensed under a Creative Commons Attribution license (CC-BY 4.0), which means that we are free to use, reproduce and distribute the article and related content if we cite the original source. We have now made this clear in the legend to Figure 4D.
Other points
Line 2, the title, LU domain, may be a good idea not to use abbreviation, use Ly6/uPAR(LU) instead?
It is not possible to avoid abbreviations as Ly6 is Ly6 antigens and uPAR is urokinase-type plasminogen activator receptor. LU domain is the accepted abbreviation for this protein domain in NCBI and we think this is better than use Ly6/uPAR.
Line 18, sounds like the multi LU domain proteins are unique to Human genome? If not, please make this clear. Good point, this has now been corrected.
Line 19, “where a dynamic….”seems excessive in giving an example of multi-LU domain protein. Please use the limited abstract space effectively to include main conclusions. Again, a good point, we were just too focused on our very recent data showing that protein flexibility of the LU is correlated to the loss of a plesiotypic disulfide bond. Nonetheless, we agree that this is too specific to be mentioned in the abstract.
Line 36, please give a reference for the mentioned highly mobile domains. Done
Line 49, are all LU domain protein genes organised in this way? And in all species? Is this organization related to exon shuffling evolution you mentioned? To the best of our knowledge, this is how the genes encoding LU domain are organized. In a very few cases the one intron separating the two exons have been deleted, but the entire gene fragment encoding the LU domain is still flanked by a symmetrical intron-exon boundary. As we self-evidently cannot state that all LU domains in all organisms have this gene structure (as this information is not available), we have modified the statement to be a “typical LU domain–encoding gene”.
The symmetrical exon-set is a prerequisite for obtaining high genetic mobility via exons shuffling, but having this gene organization does not per se imply that that particular protein has underwent exon shuffling yet. In the case of the LU domain, we conclude that it is best characterized as being a “proto-module” fulfilling all the requirements for being able to utilize this process, but only have done so in a very few instances.
Figure1, the cartoon presentation, esp. CD59, ideally turn the shadows off, so the label of the “beta F” could be more visible. Figure legend, please use references after PDB codes if there are publications associated. This applies to all the other figures/contents. We have now change the figure so that the β-strands are clearly labelled. And we have added the references to the PDB structures when available.
Line 187 and 212 are suddenly changed to functional description in the subtitles, not match to the style of others describing a certain category of proteins. Agree, there is an inconsistency here. This has now been corrected
Line 252, needs a reference This has now been added.
Line 298-300, “as the acidic…..” is excessive for the figure legend, can be moved to text? We could do that if requested, but we would prefer to keep it as it is as this information is important for the model in panel D of the space occupied by the acidic IDR of GPIHBP1. This is also one of the best characterize and essential modifications the LU domain, which has a profound biological and medical impact – this is why we have devoted space to discuss this in detail.
Line 372, what is ATF stand for? Amino-terminal domain? It seems not a standard abbreviation, may need a full description. This has now been defined.
Reviewer 2 Report
The manuscript of Julie Leth et al. is devoted to very interesting field – the evolution of three-finger proteins. Unfortunately, manuscript contains several serious shortcomings.
First, the review is very superficial, non-informative, and does not contain any new analysis of the known literature data. We are sorry that this particular reviewer has that opinion and needless to say that we disagree completely with this statement. We have tried to discuss general structural and functional data for selected LU domain proteins across different phyla, which we consider relevant‒not only focusing on the human genome or one single function. Furthermore, the fact that we discuss that human LU domain proteins (as illustrated by the first LU domain in uPAR) may have used the same evolutionary mechanism as Fry et al proposed for diversification of snake venom alpha-neurotoxins is new (our original paper on this was published only two days before this review was submitted to IJMS). This is to the best of our knowledge the first review that focuses on the relevance of the plasticity in the plesiotypic disulfide bonds in the evolution of LU domains. Other studies e.g. by Galat focused on other parameters such as hydrophobicity and sequence identity (which is very low amongst LU domains and is primarily accounted for by the consensus cysteines).
The title does not correspond to the manuscript content. We discuss both evolution and medical relevance for the LU domains when these are available, so again we do not fully understand this comment. One example is all the medical/pathological implication of genetic deficiencies, missense mutations and autoantibody syndrome in e.g. GPIHBP1, which is very new and is well received and accepted in the field of lipid metabolism. Genetic screening of GPIHBP1 is now routine when children are born with hypertriglyceridemia and we foresee that our discovery that this disease can be acquired later on life by development aoutoanti-GPIHBP1 antibodies will be incorporated in the clinics too. When data are solid we have also included examples of disease association of other LU domain–containing proteins. But there are always room for improvements and we have now included data on LY6E as proposed by reviewer 3.
There are several disappointing omissions. For example, in the chapter 2 Lypd7, Lynx2, and Lypd6 are absent in the list of human genes, although Lypd6 is mentioned several times in the manuscript. This is a clear-cut misreading from this reviewer, as we clearly stated in the manuscript that the list contain genes that werelocated in defined clusters not a complete list of all 45 known genes: Accordingly, Loughner et al. (2) found that 30 out of the 45 LU containing−proteins in the human genome are located in just four small gene clusters. These segments are located on chromosomes 6p21 (LY6G6C, LY6G6D, LY6G6F, LY6G5C, and LY6G5B), 8q24 (PSCA, LY6K, SLURP1, LYPD2, LYNX1, SLURP2, LY6D, GML, LY6E, LY6L, LY6H, and GPIHBP1), 11q24.2 (ACRV1, PATE1, PATE2, PATE3, and PATE4), and 19q13 (LYPD4, CD177, TEX101, LYPD3, PINLYP, PLAUR, LYPD5, and SPACA4). The latter gene cluster includes all proteins in the human genome known to contain multiple LU domains (3). Nonetheless, to make the list complete we have now listed the remaining genes in the end of the paragraph (note LYPD1 is LYNX2 and LYPD6B is LYPD7).
No data about the human secreted protein SLURP-2 is discussed in the paper, although its knockout also leads to skin disorders as well as mutations in SLURP-1 (Allan et al, 2016). We are fully aware of this finding as one of the co-authors on this review paper (Dr. Stephen G. Young) actually was the one who originally conducted the mouse gene ablation studies of both SLURP1 and SLURP2! There are however to the best of our knowledge no human mal de Meleda patients with defined missense mutations in SLURP2. Due to the space limitation, we cannot discuss every single one of the 45 LU domain proteins in the human genome in detail as well as making parallel discussion in other phyla. Nonetheless, as a happy compromise, we have now include a brief discussion on SLURP2 as well.
Moreover, recently, the antitumor properties of SLURP-1 and SLURP-2 were demonstrated (Lyukmanova et al., 2018). Human protein Lynx1 was studied extensively by several groups (Miwa, Morishita, Thomsen, Lyukmanova), but no data were discussed in the present review. We tried to make a selection of important LU domain-containing proteins to illustrate different functions that pertain to structural variation. In that process, we admittedly on a wrong basis excluded lynx1 as correctly pointed out by this reviewer. We have therefore included a section describing this protein, but we cannot cover all aspects of its function in such a brief and general review. There are literally hundreds of papers describing the role of uPAR in cancer invasion, metastasis and prognosis and although this is right within our own expertise, we have chosen a brief description on this relationship to keep an overall balance in the review.
It is strange to read that “ECD of type I receptors ALL comply with… LU domain signature…”. For example, toll-like receptors also belong to the family of type I receptors, but their ECDs don’t contain LU domains. This is again a clear misconception from this reviewer as this section discusses the TGF-beta receptor family only, as clearly stated in the first line of the section! This receptor family are composed of type-1 and type-2 receptor subunits, but that has no bearing on the Type 1 and Type 2 nomenclature dividing the integral membrane proteins. TLRs belong to a completely different receptor family. Nonetheless, we fully acknowledge that the sentence on page p44 in the original version was misleading and could have caused this confusion. We have therefore changed this to make it more clear. Thanks for this comment. Conclusions of the review are too brief and do bring something new for the readers. Being a review it would be strange to include an lengthy Conclusion as the various focus areas are discussed within the bulk text of the review.
Reviewer 3 Report
Comments and Suggestions for Authors
This review focuses on the very heterogeneous LU domain protein family, selecting representatives from insects, fish, snakes, and mammals.
Generally, I recommend reducing the text considerably by removing redundant listings, such as numbers of cysteines, or redundancies in alignments or structural representations (see below).
1.) disulfide bonds
For any gene family, the absence of specific cysteine bonds is mentioned separately, although the structural/functional implications are not described. This can be summarized into one paragraph, where evolutionary diversification and flexibility are discussed. As the authors mentioned in chapter 2, certain disulfide bonds are invariant in the LU domain family, whereas others were independently lost (2-3 or 7-8). In the case of uPAR, it seems that the absence of the cysteines 7-8 in the first domain can be at least partially compensated by an extension in beta strand E. The original idea motivating the writing this review was in fact to discuss the loss of plesiotypic disulfide bonds and gains of apotypic disulfide bonds across the LU domain family. A similar approach was originally pioneered for snake venom toxins by Dr Fry and colleges, but having worked with multiple LU domain-containing proteins (uPAR, C4.4A and Haldisin), where the loss of the 7-8 disulfide bond in uPAR DI plays an important role for the assembly of a high affinity binding site for uPA, we felt that a review of LU domain proteins with this as the primary focus was lacking. We do not believe that an extension of the strand E in uPAR DI compensates for the loss of the 7-8 disulfide bond, the flexibility of the LU domains in uPAR should not be viewed as isolated domains but rather as an integrated three domain assembly (Leth et al JBC 2019, and Mertens et al JBC 2012).
2.) Redundancy in figures and panels
I suggest reconsidering whether all figures are necessary. For instance, some of the alignments could be removed. Some sequences are represented twice (Bouncer). In another case, two sequences (SPT-10, GPIHBP1) are aligned not because they are closely related, but because they both have a disordered, unrelated region outside of the LU domain, where an alignment is not meaningful. In addition, providing numerous cartoon representations of three-finger fold proteins does not give deeper insights, unless specific aspects are described in the text. We agree that there is some redundancy in the figures. We have now removed Bouncer in Fig 1B, the idea with this figure was to provide an “overture” to the review showing various LU domains across different phyla. Figure 2 was to outline the well documented evolutionary trajectory of snake venom toxins, which sets the theme of the review regarding the functional diversification of this fold and the consequences of the loss of the 2-3 disulfide bond. Figure 3 was intended to support the discussion of the Bouncer isotypes (could be omitted if this reviewer finds this relevant). I consider Figure 4 very important and central as this describes a very unusual assembly of the folded LU domain and an intrinsically disordered N-terminal extension (encoded by a separate exon). Studies we have performed in the last two years have shown the functional importance of this architecture in intravascular lipolysis providing a very interesting interplay between order and disorder in the kinetics of receptor ligand interactions and in the protection from endogenous inhibitors. Admittedly, the alignment of the disordered regions is not meaningful and is not attempted either – we have therefore specified in the legend that the alignment is for the LU domain only. Figure 5 is also very essential for this review as this demonstrates the assembly of the LU domains in uPAR and the location of the deleted disulfide bonds. For all multiple LU domain proteins, the deletion of the disulfide bonds only occur in the first domain – the remaining domains maintain the ancestral 10 cysteine signature. Finally, we also think that the figure 6 with the TGF-β superfamily is relevant as this again shows a special use of the LU domain in integral membrane proteins. We therefore think that the alignments nicely supports different aspects discussed in the text.
3.) Figure with multimeric structures
It can be of interest to illustrate in a separate figure the diversity of biological assemblies, ranging from homodimers (5XWE), heterodimers of two LU-domain proteins (irditoxin, Fig 2b), hetero-4-mer (bmp-2 ligand-receptor complex, 2H62) to homo-12-mer (kappa-bungarotoxin, 1KBA). Several of these structures are already shown in the manuscript and we will prefer that they reflec the assembly of LU domains rather than complexes between LU domains and their ligands as this is an important theme in the present review.
4.) Plesiotypic/apotypic
This review is addressed to a general audience. For clarity, the authors should define the terms "plesiotypic" and "apotypic" in the introduction. Agree, this has now been introduced in the Introduction.
The word "plesiotypic" is mentioned 40 times in the text. I suggest removing some of them.
Embarrassing , we have now limited the use of this word.
5.) Secondary structure elements
Plotting a 2D structure representation of one sequence into the alignment would facilitate assigning loop regions and assessing the conservation of the structural elements. Agree, we have now included that in figure 1.
6.) PFAM conserved domains
In the PFAM database the Ly6/uPAR family is represented by the uPAR_Ly6_toxin clan (CL0117), consisting of 7 family members, each with family-specific conserved residues. A classification scheme based on these criteria would be more meaningful than the absence of specific cysteine bonds. Agree, but that would make the review similar to other reviews and would detach the focus of the review, which we chose to devote to the variety of disulfide bonding within this domains family. As alluded to previously, the importance of this focus becomes very pertinent when you work with the multi-domain members. We have now emphasized the scope of the review in the introduction.
7 .) LY6E and viral infection
To highlight the medical significance of the review, the authors could mention that LY6E and related LU family members enhance the infectivity of several RNA viruses, as shown for example by Mar et al., using an influenza A virus model system (Mar et al, Nat. comm. volume 9 (2018) PMID: 30190477, and references therein). Excellent point and very interesting reading (completely skipped our attention) – this has now been implemented in the review. Thanks for this guidance.
8.) from line 230 onwards: medaka bouncer A and B
The authors should specify that the ortholog of zebrafish bouncer is medaka bouncer B. This was shown by phylogenetic analysis by Herberg et al., and indicates that bouncer gene duplication occurred prior to speciation of the Clupeocephala (the subgroup of teleost fishes that were analyzed in this study). Medaka bouncer B protein is 40% identical to zebrafish bouncer in the mature domain, whereas bouncer A shows only 30% identity (Herberg et al). In addition, medaka bouncer A is not predicted to be GPI anchored and lacks the C-terminal transmembrane region (bigPI, TMHMM). Thus, it is unlikely that medaka bouncer A can functionally replace the GPI-anchored zebrafish bouncer (while bouncer B does so). The phylogenetic relationship would be better reflected, if the names of medaka bouncer A and B were swapped, and the order in the alignment changed accordingly. A highly relevant comment. We have now revised accordingly.
9.) line 219
Specify "zebrafish" for the chromosome for clarity. Good point